# Evaluation of Matrix Metalloproteases by Artificial Intelligence Techniques in Negative Biopsies as New Diagnostic Strategy in Prostate Cancer

**DOI:** 10.3390/ijms24087022

**Published:** 2023-04-10

**Authors:** Noemi Eiro, Antonio Medina, Luis O. Gonzalez, Maria Fraile, Ana Palacios, Safwan Escaf, Jesús M. Fernández-Gómez, Francisco J. Vizoso

**Affiliations:** 1Research Unit, Fundación Hospital de Jove, Avda. Eduardo Castro, 161, 33920 Gijón, Spain; 2Department of Anatomical Pathology, Fundación Hospital de Jove, Avda. Eduardo Castro, 161, 33920 Gijón, Spain; 3Department of Urology, Hospital Universitario Central de Asturias, Universidad de Oviedo, Avda. de Roma s/n, 33011 Oviedo, Spain

**Keywords:** tumor stroma, cancer-associated fibroblast, matrix metalloproteinases, automatic learning techniques, cancer diagnosis

## Abstract

Usually, after an abnormal level of serum prostate-specific antigen (PSA) or digital rectal exam, men undergo a prostate needle biopsy. However, the traditional sextant technique misses 15–46% of cancers. At present, there are problems regarding disease diagnosis/prognosis, especially in patients’ classification, because the information to be handled is complex and challenging to process. Matrix metalloproteases (MMPs) have high expression by prostate cancer (PCa) compared with benign prostate tissues. To assess the possible contribution to the diagnosis of PCa, we evaluated the expression of several MMPs in prostate tissues before and after PCa diagnosis using machine learning, classifiers, and supervised algorithms. A retrospective study was conducted on 29 patients diagnosed with PCa with previous benign needle biopsies, 45 patients with benign prostatic hyperplasia (BHP), and 18 patients with high-grade prostatic intraepithelial neoplasia (HGPIN). An immunohistochemical study was performed on tissue samples from tumor and non-tumor areas using specific antibodies against MMP -2, 9, 11, and 13, and the tissue inhibitor of MMPs -3 (TIMP-3), and the protein expression by different cell types was analyzed to which several automatic learning techniques have been applied. Compared with BHP or HGPIN specimens, epithelial cells (ECs) and fibroblasts from benign prostate biopsies before the diagnosis of PCa showed a significantly higher expression of MMPs and TIMP-3. Machine learning techniques provide a differentiable classification between these patients, with greater than 95% accuracy, considering ECs, being slightly lower when considering fibroblasts. In addition, evolutionary changes were found in paired tissues from benign biopsy to prostatectomy specimens in the same patient. Thus, ECs from the tumor zone from prostatectomy showed higher expressions of MMPs and TIMP-3 compared to ECs of the corresponding zone from the benign biopsy. Similar differences were found for expressions of MMP-9 and TIMP-3, between fibroblasts from these zones. The classifiers have determined that patients with benign prostate biopsies before the diagnosis of PCa showed a high MMPs/TIMP-3 expression by ECs, so in the zone without future cancer development as in the zone with future tumor, compared with biopsy samples from patients with BPH or HGPIN. Expression of MMP -2, 9, 11, and 13, and TIMP-3 phenotypically define ECs associated with future tumor development. Also, the results suggest that MMPs/TIMPs expression in biopsy tissues may reflect evolutionary changes from prostate benign tissues to PCa. Thus, these findings in combination with other parameters might contribute to improving the suspicion of PCa diagnosis.

## 1. Introduction

Prostate cancer (PCa) is the most common cancer in men; approximately 75% of the diagnosed patients are 65 or older. Therefore, considering the progressive increase in the age of the population and that the mortality rates of PCa are increasing with age, a dramatic increase in PCa incidence and mortality is expected. For all of this, an early diagnosis is key [1,2].

Despite the controversy, prostate-specific antigen (PSA) screening is generally considered for detecting PCa [3]. PSA is a serine proteolytic enzyme produced by both normal and tumoral prostatic epithelium. A serum PSA level exceeding 4 ng/mL is considered abnormal. The use of serum PSA permits the early detection of PCa and the optimization of an effective biopsy technique. Prostate needle biopsy is systematic transrectal ultrasound-guided, during which a urologist uses an ultrasound probe placed in the rectum and obtains 12 or more needle core biopsies from standard locations in the gland. If a prostate biopsy is positive for cancer, open, laparoscopic, or robotic radical prostatectomy is the definitive treatment of localized PCa.

PCa is the only solid organ malignancy that is diagnosed by such random systematic biopsies. However, the specificity of this method is limited because an elevated serum PSA concentration is not specific to carcinomas but may be due to benign conditions, such as benign prostatic hyperplasia (BPH), prostatitis, trauma, infarction, or manipulations. This standard approach is associated with disturbingly high rates of false negative diagnosis, overdiagnosis, and underdiagnosis [4,5,6]. Concerning this latter issue, the traditional sextant technique misses 15–46% of cancers [7]. Although some progress has been made in terms of improvement, such as the multiparametric MRI targeted biopsy of the prostate [6,8,9,10,11,12], which is now the recommended standard, the ideal strategy for PCa diagnosis is still to be completely defined.

Matrix metalloproteases (MMPs) have reached an extraordinary interest in cancer research due to their role in tumor invasion and metastasis, by degrading basal membrane and extracellular matrix degradation [13]. In addition, MMPs are able to impact other basic processes of tumor progression, such as inhibiting apoptosis, stimulating proliferation, or regulating cancer/related angiogenesis (for review: [14]). On the other hand, it is assumed that tissue inhibitors of MMPs (TIMPs) are multifactorial proteins also involved in the induction of cell proliferation and the inhibition of apoptosis [15]. The expression of several MMPs and TIMPs, such as MMP -2, -7, -9, -11, -13 or -14, TIMP-1, -2 or -3, has been reported and associated with the development of tumor aggressiveness and poor prognosis in PCa [16,17,18,19,20,21,22,23,24,25,26,27]. It has been also suggested that the high expression of MMPs/TIMPs by PCa compared with benign tissues could contribute to diagnosis [28].

The objective of the present work was to explore the evolutionary behavior of the expression of MMP -2, 9, 11, and 13 and TIMP-3, from benign prostate tissues to cancer, and their possible contribution to predicting PCa development. For this purpose, we analyzed especially benign biopsies from patients previous to PCa diagnosis compared with tissue samples from benign prostatic hyperplasia (BHP) and high-grade intraepithelial neoplasia (HGPIN), using machine learning techniques, and thus obtained intelligent classifiers able to predict the diagnosis of PCa.

## 2. Results

In the group of patients with PCa, the average number of biopsies before cancer diagnoses was 1.23 (range, 1–7: 14 patients underwent two biopsies, 4 underwent three biopsies, and 1 underwent seven biopsies).

Figure 1 shows some examples of immunostaining for each protein evaluated in different prostate tissues in benign or malignant prostate tissues. Immunostaining for all the proteins studied was localized predominantly in tumor cells and epithelial cells, but also in a significant percentage of stromal fibroblasts. Figure 2 and Figure 3 show the percentages of positive cases for each protein in different tissue samples.

### 2.1. Comparative Study of the Expression of Factors among Different Benign Tissues Accuracy

We found a significantly higher expression of factors by ECs and fibroblasts from benign prostate biopsies before the diagnosis of PCa, so these expressions in zones without future cancer development (C1) as in zones with future tumor (C2), compared with prostate biopsy samples from patients with BPH (C5) or HGPIN (C6) (Figure 2, Figure 3, Figure 4 and Figure 5). These findings suggest the existence of biological changes in prostate tissues preceding PCa diagnosis.

The results regarding ECs are confirmed by the classifiers. As can be seen in Table 1, ECs provide very relevant information that can be used to differentiate C1 or C2 patients concerning C5 and C6, respectively, since the accuracy is high. Indeed, C1 vs. C5 has a hit of 97.2% in FURIA and SVM and C2 vs. C6 has a hit of 98% in FURIA, for example.

Regarding fibroblasts, it can be determined that the information is also remarkable but to a lesser extent than concerning ECs. As can be seen in Table 2, C1 vs. C5 had an accuracy of 84.2% and C2 vs. C6 had an accuracy of 81.7% in FURIA, for example.

### 2.2. Evolutionary Changes in Paired Prostate Tissues from Benign Tissues to Cancer Diagnosis

To investigate the evolutionary changes of prostate tissues from benign tissue biopsies to PCa, we compared the expression of factors in biopsies before cancer diagnosis and in the prostatectomy specimen, from the same patients. For this purpose, we analyzed the sextant zones of the prostate before cancer diagnosis in two major zones: the zone without future cancer development and the zone with future cancer development. In addition, prostate zones corresponding to prostatectomy specimens were analyzed in two zones: the zone containing the tumor (tumor zone) and the zone distant to the tumor at least 1 cm (non-tumor zone) (Figure 6).

Figure 2 and Figure 3 show the expression of the studied factors in the differentiated tissues, and Figure 6 the significant differences from the several comparisons. Firstly, in benign biopsies before PCa diagnosis, we found no significant differences in MMPs/TIMP-3 expression by ECs or fibroblasts, between zones either without (C1) or with future tumor (C2). This data is confirmed by the intelligent algorithms since they obtain a hit rate of less than 75% in the C1 vs. C2 experimentation in the different study factors, ECs, and fibroblasts. Therefore, this finding suggests that the expression of these factors does not provide information about in which prostate area a subsequent cancer diagnosis will be made.

Comparing the expression of MMPs/TIMP-3 in tissue from biopsies before cancer diagnosis and in the tumor zone of the prostatectomy specimen revealed several significant differences. Malignant epithelial cells from the prostatectomy tumor site (C4) showed higher expression of MMP-2, -9, and -11, and TIMP-3 than ECs from the benign biopsy site with future cancer development (C2). Likewise, similar differences were found for the expression of MMP-9 and TIMP-3 by fibroblasts from these two zones. Thus, these differences in MMPs/TIMP-3 expression seem to correspond to molecular changes associated with the onset of cancer biology or tumor progression.

The analyses carried out with the intelligent algorithms determined that C2 and C4 can be classified/differentiated by considering the expressions of MMP-2, -9, -11, and -13, and TIMP-3 by the ECs. The FURIA algorithm showed a 91.1% accuracy. Nevertheless, when analyzing only the expressions of MMP-2, -9 and -11, and TIMP-3 in ECs, a hit of 86.2% was obtained with the FURIA algorithm implying that the relevant MMP13 expression for the discrimination between C2 and C4 samples.

However, considering MMP-9 and TIMP-3 expression by fibroblast, C2 vs. C4 has less than 75% accuracy with FURIA. If all the variables, MMPS/TIMP3, are considered, the difference between C2 and C4 samples is still unsatisfactory by intelligent algorithms, only reaching a 76.3% accuracy with FURIA.

It was also of note that our finding indicated that ECs from the non-tumor zone from prostatectomy specimens (C3) showed higher expressions of MMP-9 -11, and TIMP-3 than ECs from the zone without future cancer development from biopsy (C1). These data seem to indicate changes in the whole microenvironment from prostate associated to cancer development. This information is corroborated by the classification algorithms. Indeed, FURIA can classify samples in group C1 y C3 using the MMP-9, -11, and TIMP-3 by ECs with an accuracy of 90.8%.

Nevertheless, fibroblasts from tumor areas (C4) show higher expression of MMP-2 and TIMP-3, compared with fibroblasts from non-tumor areas from prostatectomies (C3), which indicate that more pronounced dramatic changes occur in the most intimate tumor microenvironment. Although the intelligent systems do not obtain an optimal behavior between MMP-2 and TIMP-3 expressions by fibroblasts, the data and variables are susceptible to study since the differentiation between samples C3 and C4 was 85%.

## 3. Discussion

Our results are in accordance with previous reports showing that MMPs are overexpressed in PCa in comparison with prostate benign tissues [21,28]. MMPs have a key role in cancer progression by positively affecting several basic processes, such as invasion metastasis, angiogenesis, or proliferation (for review: [14,29]). Consequently, MMP expression was also associated with tumor progression and poor prognostic in PCa [21,23]. The present study suggests an alteration in the expression of these factors in situations previous to the diagnosis of PCa. Thus, these findings led us to consider that these changes may be in relation to the presence of the cancer microenvironment, although the result of the biopsy was negative, or as a consequence of microenvironment changes related to malignant transformation of the prostate epithelium.

These early changes can be expected to occur in HGPIN. It is generally accepted that HGPIN is a preneoplastic lesion which initiates organ-confined prostate adenocarcinoma. It is assumed that HGPIN initiates a combination of cellular events that trigger a cascade of genomic instability [30]. In addition, HGPIN shows histological changes such as nuclear atypia, loss of cellular polarity, focal dysplasia [31], loss of neuroendocrine and secretory differentiation, nuclear and nucleolar abnormalities, neovascularity, increased proliferative potential and genetic instability with the variation of DNA content [32]. However, the basal lamina remains intact during PIN, which alteration is caused by the involvement of MMPs. Therefore, our data seems to indicate that overexpression of MMPs may be a more definitive event associated with tumor development. It is also noteworthy that these findings related to tumors are affected not only by epithelial cell phenotype but also that of stromal fibroblasts, which is in line with the importance assigned to this tissue compartment in prostate development [33,34,35]. Thus, our data indicate that changes in ECs and reactive stroma, which are composed mainly of carcinoma-associated fibroblasts, are initiated during early PCa development.

In the present study, we also evaluated the evolutionary changes in the expression of MMPs/TIMP-3, both in the area where the tumor arose and in those far away from them in the previous benign biopsy and the prostatectomy specimen (Figure 6). Most factors were overexpressed by epithelial malignant cells compared with the benign epithelial cells from the corresponding zone with a benign previous biopsy. Nevertheless, higher expressions of some factors were observed by stromal cancer-associated fibroblasts compared with fibroblasts from previous biopsy (MMP-9 and TIMP-3) or to fibroblasts from non-tumor zones of prostatectomy (MMP-2 and TIMP-3). On the other hand, it was also remarkable that we found high expressions of several factors by ECs from non-tumor areas from prostatectomies (MMP-9 and 11, and TIMP-3) compared with these same cell types from previous benign biopsies in the zone where the tumor arose. Thus, the latter data suggest that possible tumor-derived microenvironment alterations may involve the whole prostate gland. This may be a differential fact of PCa due to its development in an organ-confined gland, which might have importance to better diagnose PCa.

We chose these MMPs and TIMP-3 for their importance in tumor progression. MMP-2 (gelatinase A) and MMP-9 (gelatinase B) are related to tumor invasion and metastasis by their special capacity to degrade the type IV collagen found in basement membranes [36,37]. Previously, MMP-9 has been found overexpressed in stromal cells from PCa and MMP-9 with biochemical recurrence [23]. MMP-11 (also known as stromelysin-3) has relatively weak proteolytic potential compared with other MMPs [36]. However, whereas most MMPs are secreted as proenzymes that need extracellular activation, MMP-11 is processed intracellularly and secreted as an active enzyme, suggesting that MMP-11 may have a unique role in tumor development and progression [38,39,40]. In addition, it has been proposed that, although in tumorigenesis induced by MMP-11 cancer cell proliferation was not increased, cancer cell death through apoptosis and necrosis was decreased, indicating that the function of MMP-11 is to promote cancer cell survival in the stromal environment [41]. Interestingly, MMP-11 expression was associated with tumor progression [21,35] and castration resistance in PCa [35]. Regarding TIMP-3, its expression by fibroblasts was associated with a higher Gleason score [22]. Although TIMPs inhibit MMPs, the multifactorial proteins are also involved in the proliferation and the inhibition of apoptosis [42].

On the other hand, we used machine learning techniques to improve a differentiable classification among the patient populations and prostate tissues. Diagnostic prediction of diseases, including cancer, is a field addressed by machine learning and artificial intelligence [43,44,45,46,47,48,49]. The importance of classifying cancer patients and the detection of key features from complex datasets, especially those that depend on complex proteomic and genomic measurements, led to the development of machine learning techniques aiming to model the progression of cancers [50]. A variety of these techniques, including Fuzzy Rule-Based Systems (FRBS) [51], have been widely applied in cancer research for the development of predictive models, resulting in effective and accurate decision-making.

Machine learning is a branch of artificial intelligence that employs a variety of statistical, probabilistic, and optimization techniques that allows computers to “learn” from past examples and to detect hard-to-discern patterns from large, noisy, or complex datasets [50]. As a result, machine learning is frequently used in cancer diagnosis and detection. A recent study evidenced that by using the same clinical parameters, machine learning techniques performed better than the European Randomized Study of Screening for Prostate Cancer (ERSPC) risk calculator (ERSPC-RC) or PSA density in clinically significant PCa predictions and could avoid up to 50% unnecessary biopsies [52]. In the present study, we achieved greater than 95% accuracy, considering ECs, being slightly lower when considering fibroblasts, to classify patients as at risk of PCa diagnosis.

There are some limitations in the present study. The study is retrospective but it was necessary to be so to obtain these promising data, which will allow a future prospective study to be planned. Another limitation is that the size of the cylinder analyzed is much smaller than the prostatic sextant, although the impact was minimized by choosing a non-tumor area away from the tumor site.

In summary, our data suggested that the expression of MMPs/TIMPs in prostate biopsies, and in combination with other parameters, might contribute to improving the suspicion of PCa diagnosis. This is because there is a percentage of false negatives in needle biopsies in PCa in which cancer cells are not detected. Thus, the possible presence of benign ECs or fibroblasts showing positive immunostaining for several MMPs/TIMPs may add useful information for the early detection of a phenotype that can be classified as high risk of PCa development. Even though it is evident that the use of machine learning methods can improve our understanding of cancer progression, an appropriate level of validation is needed for these methods to be considered in everyday clinical practice.

## 4. Materials and Methods

### 4.1. Patients

A retrospective study was conducted enrolling patients at Hospital de Jove (Gijón, Spain) and Hospital San Agustin (Avilés, Spain), both from Spain. The main group of patients included in the present study was 29 patients (age range, 54–73 years) with PCa diagnosis after at least one initial negative biopsy. All patients underwent prostate biopsies due to abnormal serum PSA levels (>4 ng/mL), abnormal findings on digital rectal examination (DRE), and abnormal findings by transrectal ultrasound (TRUS).

The PSA serum levels were determined using the ‘Elecys’ immune-assay tests (Roche Diagnostic GmbH, Mannheim, Germany). Transrectal prostate biopsies were guided by ultrasonography (Type 2202, BK medical, Herlev, Denmark; in two-dimensional planes (sagittal and axial)). Twelve cores were obtained in each patient, identified according to their location: base, mid, and apex, from the left and right side. In patients with a benign result, PSA concentrations were evaluated at months 3 and 6 and then every 6 months or 1 year. A transrectal prostate biopsy was performed in patients with a PSA concentration >4.0 ng/mL or a biopsy finding of an atypical prostatic gland. After the diagnosis of PCa, patients underwent radical retropubic prostatectomy, the specimen was identified according to the location: base, mid, and apex, from the left and right side. Tumors were staged according to the 1992 TNM classification [53].

In the present study, two other patient populations were included corresponding to 45 patients with histological diagnosis of BPH (age range, 44–85 years) and 18 patients with HGPIN (age range, 54–70 years), some of which were previously included in our preliminary studies on the expression of MMPs and TIMPs in prostate benign tissues [21,28].

All patients were treated according to the guidelines used in our institutions. The study adhered to national regulations and was approved by our institution’s Ethics and Investigation Committee.

### 4.2. Immunohistochemical Analysis

The histological material used in this study was obtained from samples of needle biopsy and also from radical retropubic prostatectomy specimens from patients who developed PCa, and from adenomectomy specimens from patients with BPH and HGPIN.

All prostate specimens were routinely formalin-fixed paraffin-embedded (FFPE). Histopathological representative (tumor/no tumor) areas were defined on hematoxylin and eosin-stained sections by an expert pathologist (L.O.G). Immunohistochemistry was carried out on a 5 µm tissue section using a TechMate TM50 autostainer (Dako, Glostrup, Denmark). Antibodies for MMPs and TIMPs were obtained from ThermoFisher Scientific (Waltham, MA, USA). The dilution for each antibody was 1/25 for MMP-13 (MA5-14238); 1/100 for MMP-2 (MS-806P1), MMP-9 (MA1-12894) and TIMP-3 (PA1-38778 and 1/500 for MMP-11 (MA5-32285). All the dilutions were made in Antibody Diluent (Dako) and incubated for 1 h (MMP-9), 2 h (MMP-2, MMP-11, and MMP-13), or overnight (TIMP-3) at room temperature. Tissue sections were deparaffinized in xylene, and then rehydrated in graded concentrations of ethyl alcohol (100, 96, 80, 70%, then water). To enhance antigen retrieval only for some antibodies, tissue sections were treated in a PT-Link^®^ (Dako) at 97 °C for 20 min, in citrate buffer pH 6. 1 for MMP-9 and TIMP-3, or Tris-EDTA buffer pH 9 for MMP-2 and MMP-13, and then washed in phosphate-buffered saline (PBS). Endogenous peroxidase activity was blocked by incubating the slides in a peroxidase-blocking solution (Dako) for 5 min. The EnVision Detection Kit (Dako) was used as the staining detection system. Sections were counterstained with hematoxylin, dehydrated with ethanol, and permanently coverslipped.

For each antibody preparation studied, immunoreactivity for epithelial cells, tumor cells, and fibroblast stromal cells was determined. A case was considered as positive for each cell type if at least 10% of considered cells showed immunostaining for a specific antibody. Due to their small size, stromal cells were considered negative (−/0) or positive (+/1) for each protein immunostaining. However, epithelial and cancer cells were classified as having weak (+/1), moderate (++/2), or strong (+++/3) immunostaining. Each evaluated field (×40 objective lens, 5 fields evaluated) contained at least ten evaluated cells, as it was established in previous studies [54,55]. Stromal cells were distinguished from cancer cells based on cell size (the latter cells are larger). Stromal cell subsets were distinguished primarily by morphology (fibroblast-like cells are spindle-shaped cells, whereas MICs are small round cells). Additionally, whereas cancer cells are arranged forming either acinar or trabecular patterns, stromal cells are scattered throughout the tissue. To confirm that the morphology described was in accordance with the cell type, specific markers were used in some tissue sections to distinguish cancer cells (Cytokeratin AE1/AE3), cancer-associated fibroblasts (α-SMA), and mononuclear inflammatory cells (CD45Ro).

### 4.3. Description of Data Sets

Artificial intelligence can perform classification tasks through intelligent systems where the expert’s knowledge will be necessary and will be determined by data. In this case, the data are the values of positive or negative expression of MMP-2, -9, -11, and -13, and TIMP-3 by each studied cell type (epithelial/tumor cells and fibroblasts) from the different patients included in the study. Machine learning techniques, a branch of artificial intelligence, comprise supervised and unsupervised learning [56]. In supervised learning, a labeled set of training data is used and in unsupervised no labeled examples are provided. In this study, we are faced with supervised learning and, therefore, with a classification problem since each data set (MMPS and TIMP-3 values for each patient) is classified in a specific class, see Table 3 and Figure 7.

Protein expression (MMP-2, -9, -11, and -13, and TIMP-3 expression) were collected and grouped into different datasets depending on the analysis to be carried out. For example, pooled data from BPH (C5) or HGPIN (C6) patients with data from patients with a benign prostate biopsy before PCa diagnosis (C2), allowed to predict whether a patient would be diagnosed as C5, C6, or C2 and, therefore, whether or not they would have PCa. In addition, each of these groups of datasets was grouped according to the target classes (Table 3) and the cell types to be studied. Therefore, for each cell type, there was a dataset of the patients in the different classes.

In the pre-experimentation phase, the datasets undergo a division into two different sets:(a)Training dataset: the data were used to train the algorithm to obtain the relevant and coherent knowledge and information capable of discriminating the input breaking down data (MMP-2, -9, -11, -13, and TIMP-3 expression);(b)Test dataset: the data were used to determine whether the behavior and knowledge provided by the intelligent system are adequate, through the corresponding evaluation by the degree of success or accuracy, and thus verify the effectiveness of the said algorithm.

The experimentation on each dataset uses the bootstrap sampling technique [57], which consists of breaking down the dataset or experiment into 100 training sets and 100 test sets. This way, each experiment is executed 100 times to study the behavior.

### 4.4. Data Analysis and Intelligent Algorithms: Description and Evaluation

Differences in percentages were calculated with the χ2-test. The SPSS 25.0 software was used for all calculations (SPSS Inc., Chicago, IL, USA).

The classification algorithms selected in this study are FURIA (Fuzzy Unordered Rule Induction Algorithm) [58,59], XGboost [60], SVM (Support vector machine) [61], Deep Learning [62], and Logistic Regression [63].

In the field of artificial intelligence, the confusion matrix, obtained in the classification of a dataset, is determined by rows and columns. Each column represents the number of predictions for each class, and the rows determine the actual class for each of the data. In two-class problems, the confusion matrix divides the result of a classification problem into four different categories, see Table 4: true positives (TP), true negatives (TN), false negatives (FN), and false positives (FP).

The training and test dataset is defined and evaluated in terms of accuracy (Acc) [49,50]
(1)Acc=1−Err
where
(2)$$Err=\frac{FP+FN}{VP+VN+FP+FN}$$

## Figures and Tables

**Figure 1 ijms-24-07022-f001:**
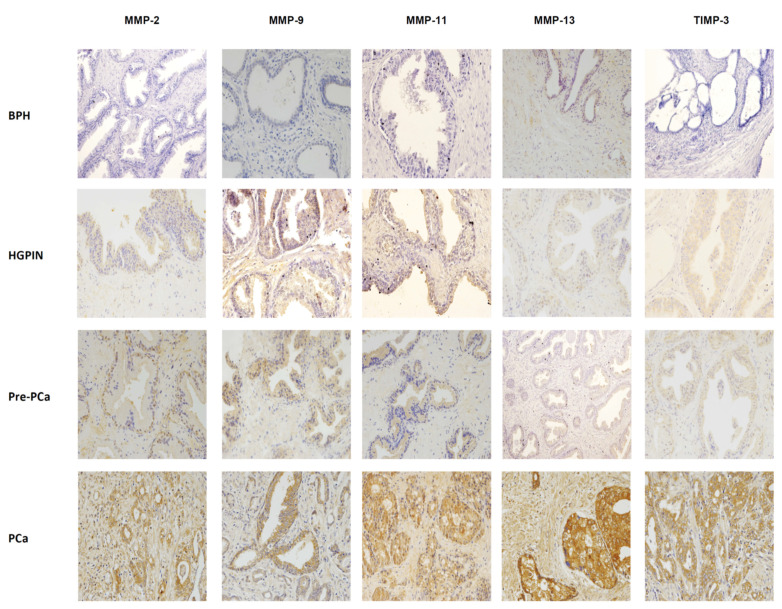
Immunohistochemical staining of MMP-2, -9, -11, -13, and TIMP-3 in benign (benign prostatic hyperplasia (BHP), high-grade prostatic intraepithelial neoplasia (HGPIN)) and malignant prostate tissues, before (Pre-CaP) and after prostate cancer (PCa) diagnosis. Magnification: ×200.

**Figure 2 ijms-24-07022-f002:**
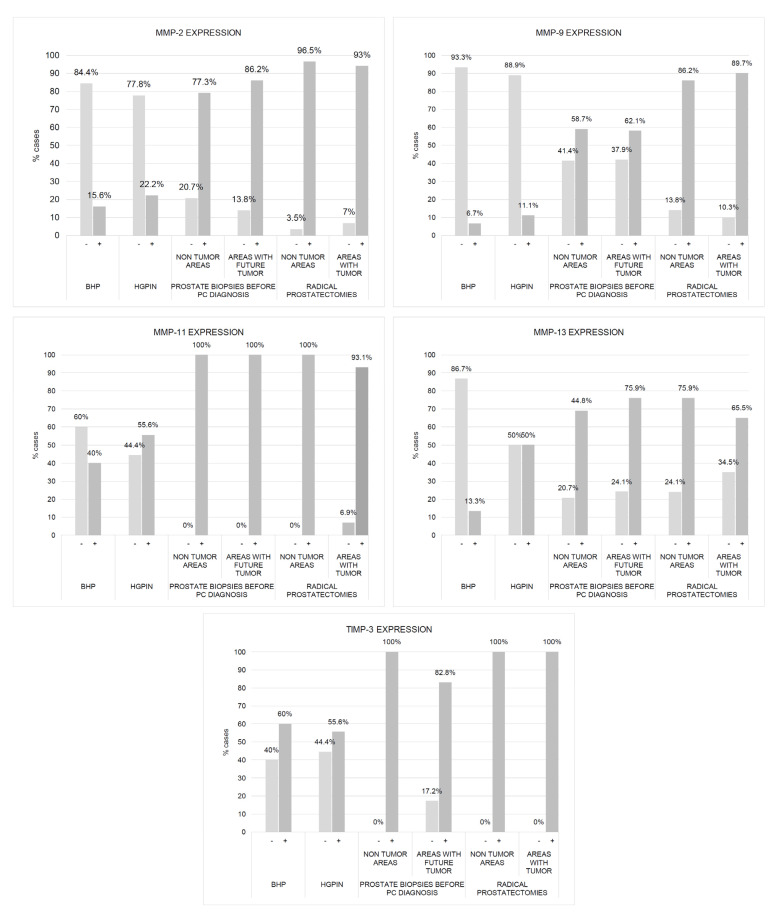
Expression of MMPs and TIMP-3 by epithelial cells from benign prostatic hyperplasia (BHP), high-grade prostatic intraepithelial neoplasia (HGPIN), benign prostate biopsies before the diagnosis of prostate cancer (non-tumor zones and zones with future tumor), and from paired zones from prostatectomies by prostate cancer (non-tumor zone and zone with tumor). The values represent the percentages of cases. Columns represent the number of cases of each series.

**Figure 3 ijms-24-07022-f003:**
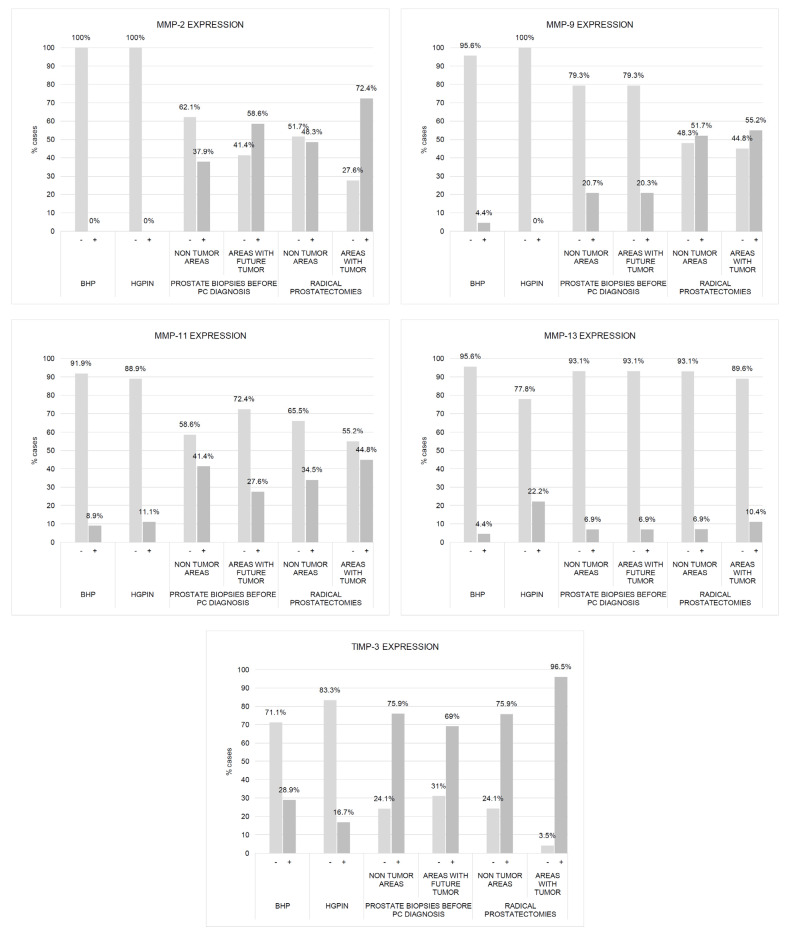
Expression of MMPs and TIMP-3 by fibroblasts from benign prostatic hyperplasia (BHP), high-grade prostatic intraepithelial neoplasia (HGPIN), benign prostate biopsies before the diagnosis of prostate cancer (non-tumor zones and zones with future tumor), and from paired zones from prostatectomies by prostate cancer (non-tumor zone and zone with tumor). The values represent the percentages of cases. Columns represent the number of cases of each series.

**Figure 4 ijms-24-07022-f004:**
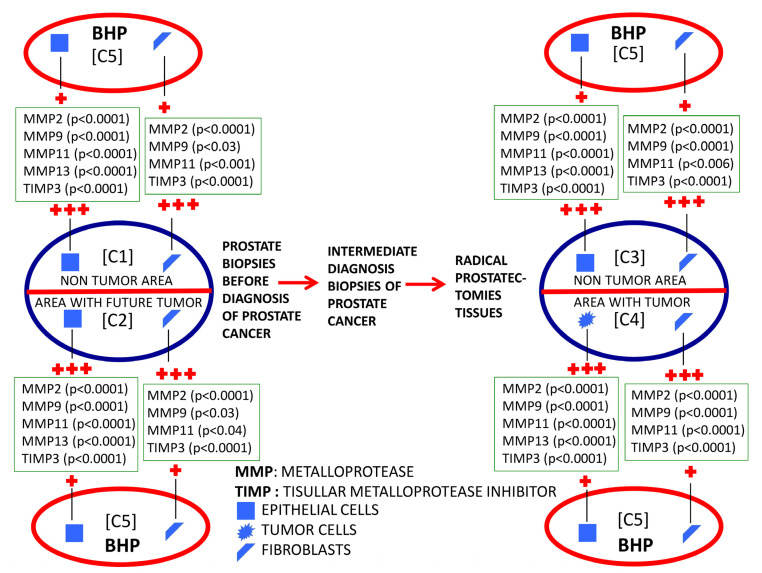
Schematic comparative study of the expression of MMPs and TIMP-3 by epithelial cells and by fibroblasts from benign prostatic hyperplasia (BHP), benign prostate biopsies before the diagnosis of prostate cancer (non-tumor zones and zones with future tumor), and from paired zones from prostatectomies by prostate cancer (non-tumor zone and zone with tumor). +: lower expression; +++: higher expression.

**Figure 5 ijms-24-07022-f005:**
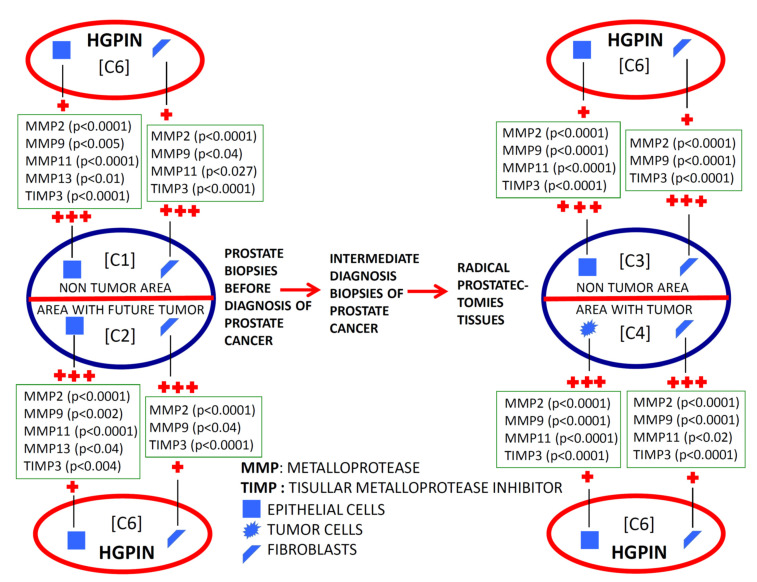
Schematic comparative study of the expression of MMPs and TIMP-3 by epithelial cells and by fibroblasts from high-grade prostatic intraepithelial neoplasia (HGPIN), benign prostate biopsies before the diagnosis of prostate cancer (non-tumor zones and zones with future tumor), and from paired zones from prostatectomies by prostate cancer (non-tumor zone and zone with tumor). +: lower expression; +++: higher expression.

**Figure 6 ijms-24-07022-f006:**
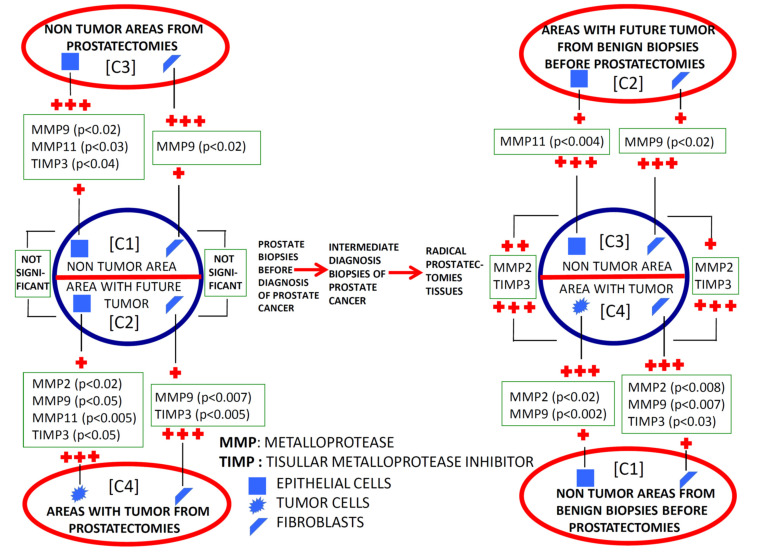
Schematic comparative study of the expression of MMPs and TIMP-3 by epithelial cells and by fibroblasts from benign prostate biopsies before the diagnosis of prostate cancer (non-tumor zones and zones with future tumor), and from paired zones from prostatectomies by prostate cancer (non-tumor zone and zone with tumor). +: lower expression; ++: middle expression; +++: higher expression.

**Figure 7 ijms-24-07022-f007:**
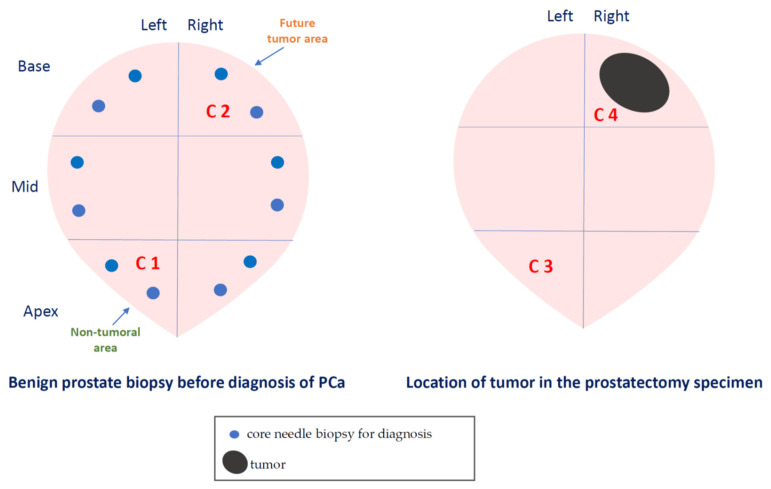
Schematic representation of the studied areas of the samples of patients with PCa.

**Table 1 ijms-24-07022-t001:** Percentage of accuracy for the different algorithms considering MMPs/TIMP-3 expression by epithelial cells.

Epithelial Cells	SVM	LR	DeepL	xbgoost	FURIA
C1 vs. C5	97.2%	95.9%	71.6%	95.9%	97.2%
C2 vs. C5	95.9%	91.8%	78.3%	93.2%	96.0%
C1 vs. C6	95.7%	93.6%	72.3%	95.7%	97.6%
C2 vs. C6	93.6%	87.2%	61.7%	82.9%	98.0%

**Table 2 ijms-24-07022-t002:** Percentage of accuracy for the different algorithms considering MMPs/TIMP-3 expression by fibroblasts.

Fibroblasts	SVM	LR	DeepL	xbgoost	FURIA
C1 vs. C5	67.5%	71.6%	62.1%	78.3%	84.2%
C2 vs. C5	83.7%	77.0%	64.8%	85.1%	87.8%
C1 vs. C6	78.7%	80.8%	57.4%	85.1%	88.5%
C2 vs. C6	78.7%	68.0%	55.3%	74.4%	81.7%

**Table 3 ijms-24-07022-t003:** Patient groups.

Benign prostate biopsy before the diagnosis of PCa Class 1: Zone without future cancer development (C1) Class 2: Zone with future tumor (C2)
Positive biopsy: PCa Class 3: Non-tumor area from prostatectomy (C3) Class 4: Tumor area from prostatectomy (C4)
Class 5: Benign prostatic hyperplasia (BPH) (C5)
Class 6: High-grade prostate intraepithelial neoplasia (HGPIN) (C6)

**Table 4 ijms-24-07022-t004:** Confusion matrix.

		Prediction	
		Positives	Negatives
True	Positives	VN	FN
	Negatives	FP	VN

## Data Availability

Not applicable.

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
