# Peer review of "Evaluation of Matrix Metalloproteases by Artificial Intelligence Techniques in Negative Biopsies as New Diagnostic Strategy in Prostate Cancer"

_ijms, 2023, doi:10.3390/ijms24087022_

Round 1
Reviewer 1 Report
With great interest I reviewed the manuscript by Eiro et al. The authors explored the expression of several matrix metalloproteases (MMPs) and tissue inhibitors of MMPs (TIMPs) in benign prostate biopsy tissue from patients previous to prostate cancer diagnosis, samples from benign prostatic hyperplasia and high-grade intraepithelial neoplasia. The aim was to gain insights into evolutionary behavior of MMPs and TIMPs on the path from benign tissue to prostate cancer.
In general, the manuscript is concise and the analyses are sound.
However, there are some minor issues that need to be clarified.
Introduction:
Reading the authors' statement on prostate biopsy, one might think that randomized sextant biopsy is still the standard. The multiparametric MRI targeted biopsy of the prostate is referred on, but not mentioned as the now recommended standard (e.g., EAU Guidelines). This issue should be clarified.
Methods, Discussion:
The authors compare the expression of the targets explored in tissue from patients before and after diagnosis of prostate cancer. It is not really clear how it is ensured that truly corresponding regions have been compared. Since randomized rather than targeted biopsies were performed in this study, it is extremely difficult to make an exact assignment in view of the possibly very small tumor foci. The procedure should be explained again in more detail and also critically discussed. The authors should also reconsider whether the study is not associated with further limitations. So far, no limitations have been mentioned.
Author Response
- Introduction: Reading the authors' statement on prostate biopsy, one might think that randomized sextant biopsy is still the standard. The multiparametric MRI targeted biopsy of the prostate is referred on, but not mentioned as the now recommended standard (e.g., EAU Guidelines). This issue should be clarified.
As suggested by the reviewer, we state that multiparametric MRI targeted biopsy of the prostate is the recommended standard. (Introduction section, page 2, line 68).
- Methods, Discussion: The authors compare the expression of the targets explored in tissue from patients before and after diagnosis of prostate cancer. It is not really clear how it is ensured that truly corresponding regions have been compared. Since randomized rather than targeted biopsies were performed in this study, it is extremely difficult to make an exact assignment in view of the possibly very small tumor foci. The procedure should be explained again in more detail and also critically discussed. The authors should also reconsider whether the study is not associated with further limitations. So far, no limitations have been mentioned.
Following the reviewer's suggestion, we have indicated that all prostate samples were identified according to their location and a new figure has been added to illustrate this (Material and methods section, page 11, lines 302-303 and 307-308).
Also, the limitations of the study have been described in the discussion section (page 11, lines 276-280).
Thank you for your evaluation.

Reviewer 2 Report
Overall well presented.
Hypothesis is that an MMP signature can predict the likelihood of a missing cancer diagnosis is relevant and reflects the weaknesses of prostate biopsy results .
Methodology is clear and based on a retrospective review of biopsies.
Conclusions are correct.
Minor typos
Line 51: ‘despite controversial’ should read’ despite controversy’
Line 61: Phrase ‘for it’ not relevant
overall well presented and with a clear statistical plan. Results presentation is high quality with graphs and figures showing applications of the MMP signature.
Author Response
Reviewer 2
Line 51: ‘despite controversial’ should read’ despite controversy’
Line 61: Phrase ‘for it’ not relevant
Typos have been corrected. Thank you for your evaluation.
